# Current and Future Spatial Distribution of the *Aedes aegypti* in Peru Based on Topoclimatic Analysis and Climate Change Scenarios

**DOI:** 10.3390/insects16050487

**Published:** 2025-05-02

**Authors:** Alex J. Vergara, Sivmny V. Valqui-Reina, Dennis Cieza-Tarrillo, Candy Lisbeth Ocaña-Zúñiga, Rocio Hernández, Sandy R. Chapa-Gonza, Erick A. Aquiñivin-Silva, Armstrong B. Fernández-Jeri, Alexandre Rosa dos Santos

**Affiliations:** 1Instituto de Investigación, Innovación y Desarrollo para el Sector Agrario y Agroindustrial (IIDAA), Facultad de Ingeniería y Ciencias Agrarias, Universidad Nacional Toribio Rodríguez de Mendoza de Amazonas, Calle Higos Urco 342—Ciudad Universitaria, Chachapoyas 01000, Peru; sivalre2001@gmail.com (S.V.V.-R.); merly.vasquez@untrm.edu.pe (R.H.); sandyrubithchapagonza@gmail.com (S.R.C.-G.); erick.auquinivin@untrm.edu.pe (E.A.A.-S.); armstrong.fernandez@untrm.edu.pe (A.B.F.-J.); 2Departamento de Ciencias Forestales, Escuela de Ingeniería Forestal y Ambiental, Universidad Nacional Autónoma de Chota, Jr. José Osores Nro. 418, Chota 06121, Peru; daciezat@gmail.com; 3Instituto de Investigación en Ciencia de Datos (INSCID), Universidad Nacional de Jaén, Carretera Jaen—San Ignacio Km. 24, Sec. Yanayacu, Jaén 06801, Peru; candy.ocana@unj.edu.pe; 4Centro de Ciências Agrárias e Engenharias, Federal University of Espírito Santo (UFES), Rua Alto Universitário, Alegre 29500-000, ES, Brazil; alexandre.r.santos@ufes.br

**Keywords:** spatial distribution, epidemic, MaxEnt, dengue, climate change

## Abstract

The geographic expansion of the *Aedes aegypti* mosquito depends on climatic variables and its population dynamics. However, human intervention has expanded its distribution, and surveillance and disease control are key to avoid considerable impacts on public health. This research uses geo-referenced data of *Ae. aegypti* findings in regions of Peru and climatic data to analyze the probability of distribution of the species. The results highlight the need to implement health measures in areas where dengue is most likely to occur in order to mitigate potential public health problems. Based on projected future climate scenarios, it is anticipated that areas with a high probability of Ae. aegypti distribution will experience a large expansion.

## 1. Introduction

Climate change entails shifts in long-term weather patterns, with significant implications for public health. Among these implications is the expansion of the geographic range of several human infectious diseases transmitted by vectors such as mosquitoes [1]. Mosquitoes possess the capacity to transmit a diverse array of parasites and pathogens responsible for serious diseases [2]. *Aedes* spp. mosquitoes are among the most important vectors of etiological agents [3], and their global distribution has garnered significant attention due to their crucial role as biological vectors for several infectious diseases [4] including dengue, chikungunya, yellow fever, and Zika virus, among others [5].

Incidence rates of *Ae. aegypti* have doubled every decade over the past 30 years as a result of increased urbanization, global mobility, and climate change [6,7]. During this period, *Ae. aegypti* has adapted to urban environments near humans; however, evidence suggests that it is also becoming established in peri-urban and rural regions of South America [8,9], where environmental conditions provide an ideal setting for the maintenance of viruses and the periodic emergence of epidemic strains [6]. The *Ae. aegypti* mosquito is the principal vector of dengue, and the incidence of dengue has risen markedly, with climate change emerging as one of the main factors exacerbating disease transmission. Among the critical climatic variables associated with dengue transmission, temperature, precipitation, and relative humidity are of paramount importance. Current estimates indicate an annual incidence of between 50 and 100 million dengue infections in tropical and subtropical regions [10] and worldwide, with estimated infections per year exceeding the 390 million, of which 96 million manifest clinically [11,12,13]. Additionally, studies suggest that climate and human behavior interact to influence the population dynamics of the dengue virus and its vector (Ae. aegypti); however, the relative effects of these variables are contingent upon local ecological and social contexts [14].

Warmer air and water temperatures can increase larval development rates [6]. The mosquito life cycle includes an aquatic phase, requiring bodies of water for oviposition and for completing the larval and pupal stages. Thus, the availability of water constitutes a critical determinant for oviposition and the successful development of immature stages [7]. *Ae. aegypti* typically lays its eggs in rainwater, often within artificial containers, where the water exhibits very low hardness, similar to that of reverse osmosis water [8]. *Ae. aegypti* eggs are capable of hatching within minutes upon contact with water; moreover, research has shown that eggs can remain viable for over a year under dry conditions. Consequently, water storage practices play a crucial role in controlling their reproduction [9]. Temperature, in turn, affects the development, growth rate, and metabolic activity of mosquitoes [10]. At lower temperatures, *Ae. aegypti* activity decreases significantly, with 12 °C representing the threshold below which development cannot occur [11]; conversely, at higher temperatures, mosquito activity increases, although extreme temperatures above 35 °C also limit its flight and feeding activities [12].

Dengue is a systemic viral infection transmitted among humans [13], with clinical manifestations ranging from mild dengue fever to more severe forms such as dengue hemorrhagic fever or dengue shock syndrome [14]. Notably, for certain patients, dengue poses a significant risk of mortality [13].

In 2023, the Pan American Health Organization (PAHO) reported the highest number of dengue cases in the Region of the Americas, totaling 4,565,911 cases, which included 7653 severe cases (0.17%) and 2340 deaths (case fatality rate of 0.051%). This surpassed the previous record observed in 2019 (3.1 million cases) by more than 1 million additional cases. The number of reports continued to increase until February 2024, during which 673,267 cases of dengue were reported, comprising 700 severe cases (0.1%) and 102 fatalities (0.1%). The 11 countries and territories experiencing heightened case counts include Argentina, Brazil, Colombia, Costa Rica, Guatemala, Guadeloupe, French Guiana, Martinique, Mexico, Paraguay, and Peru [15].

Dengue is endemic in Peru [16] and classified by the country’s Ministry of Health as a re-emerging disease [17]; more than half of the population of Peru is at risk of infection, and due to the country’s geography, the country is particularly sensitive to the effects of the El Niño Southern Oscillation (ENSO) climatic phenomenon (Dostal et al., 2022). This phenomenon plays an important role in the occurrence of infectious disease outbreaks, and their impacts on public health [18,19].

Dengue virus (DENV) includes four serotypes, DENV-1, DENV-2, DENV-3 and DENV-4. DENV causes a febrile illness (dengue fever) that can be classified as either non-severe or severe, and in certain cases, it can be fatal [20]. All four serotypes have been present in Peru since the initial emergence of the disease; however, the occurrence and prevalence of specific serotypes and their associated genotypes have varied over time [21]. In 2024, the circulation of DENV-3 was identified in the regions Lima, Loreto, San Martín, Piura, Cajamarca, Amazonas, Ancash, Ica, Callao, Ayacucho, Huánuco and Ucayali, where previously only the presence of DENV-1 and DENV-2 had been reported.

*Ae. aegypti* was first detected in Peru in 1852 and progressively became established along the northern and central Peruvian coast [22,23]. By 1905, its presence had extended as far as the port of Callao-Lima, and by 1938, it had been established in 191 localities across 11 departments [23]. Following multiple control programs, *Ae. aegypti* was eradicated from Peru in 1956 [17,22]. However, it was detected again in 1984, in the Amazonian city of Iquitos [24]; subsequently, its distribution area expanded, reaching from Tumbes to Casma (Ancash) within two years; by 2001, 12 regions, including Sullana and Pariñas (Piura), Trujillo and El Porvenir (La Libertad), and Jaén (Cajamarca), were affected by dengue outbreaks. In 2005, Lima reported for the first time numerous autochthonous cases of dengue [25]. The spread of dengue in Peru was remarkable between 2005 and 2011, with outbreaks reported in 269 districts in 18 regions; in 2015, the epidemic spread to Ica. As of July 2023, cases had been reported in all regions of Peru, with the highest numbers recorded in Piura (67,697), Lima (32,009), Lambayeque (28,235) and La Libertad (20,289) [16]. According to the Pan American Health Organization (PAHO), as of epidemiological week 6 of the year 2024, Peru had already reported 17,140 cases of dengue.

Understanding the geographic distribution and burden of *Ae. aegypti* is crucial for assessing its global impact on disease burden and dengue-related mortality [13]. In response to these challenges, geospatial modeling has emerged as a valuable tool to solve this problem; by leveraging geographic information systems and biologically relevant climatic indicators [26], it is possible to characterize the distribution patterns for various species, including *Aedes* mosquitoes, mapping their potential distribution range [27]. Species spatial modeling generally uses two types of algorithms: those that use both presence and absence data to predict species distribution, and those based solely on species presence [4,28]; among the latter, the MaxEnt (Maximum Entropy) algorithm is widely used for species distribution modeling [29,30,31] and disease vector prediction [32]. Species distribution models require the integration of ecologically relevant predictor variables specific to the species under investigation [33]. These models utilize background data to account for environmental variations in space [34], thereby estimating the most probable geographic distribution for a species by calculating the probability of occurrence [28].

The use of MaxEnt for species distribution modeling is important in our study as it permits the incorporation of diverse variables such as topography and climate. Previous research has established associations between dengue incidence and environmental factors, emphasizing the significance of climatic and landscape variables in determining *Aedes* mosquito abundance [35]. These studies have also employed MaxEnt to predict the influence of climate change on the geographic distribution of *Ae. aegypti*, using presence records and bioclimatic variables to generate maps of environmental suitability for the mosquito vector’s current geographic distribution, and, once the climatic variables influencing distribution are determined, to estimate the potential redistribution under future climate scenarios [36,37]. A modeling study in Colombia showed that the Caribbean and Andean regions exhibit a high probability of *Ae. aegypti* distribution; furthermore, it was determined that there are approximately 140,612.8 km^2^ of areas with possible vector presence, although this area is projected to decrease by more than 30% in the future [38]. In another study, Kraemer et al. [39] mapped the global distribution of the vectors *Ae. aegypti* and the geographic determinants of their ranges, finding that the distributions of *Aedes* are the widest ever recorded, and now spread over all continents, including North America and Europe.

Worldwide, the spatial distribution of *Aedes aegypti* has been studied using diverse methodologies. For example, Liu et al. [40] conducted a study in mainland China modeling the arbovirus vectors *Ae. aegypti*, employing documented occurrence records along with environmental variables to estimate the relationship between these factors using statistical algorithms. In Mexico, Candelario-Mejía et al. [41] modeled *Ae. aegypti* across various localities within the Mexican Republic, determining the species’ potential niche using a database composed of occurrence records. They incorporated referenced points and bioclimatic variables into the MaxEnt 3.3.3 algorithm, with the quality of predictions evaluated through the Receiver Operating Characteristic (ROC) technique. Furthermore, Romero et al. [42] employed fuzzy logic to assess the biogeographic risk of dengue in South America.

Previous research, such as that conducted by Jácome et al. [43], has identified various factors such as climatic, demographic, economic, and social characteristics contributing to the proliferation of the *Ae. aegypti* dengue vector. Additionally, Kraemer et al. [44] highlighted the expanding distribution of *Aedes* across continents, driven by rising global temperatures associated with climate change. Consequently, it is imperative to quantify new *Aedes* invasions to mitigate the escalating risk to human health worldwide. However, despite this urgency, few studies, including those by Kamal et al. [45], have comprehensively assessed the influences of climate change on the spatial distribution patterns and abundance of these crucial vectors, particularly utilizing the most recent climate scenarios.

Given the increasing challenges in dengue prevention and control, it is essential to obtain a comprehensive understanding of the distribution of the mosquito vector. Although the dengue epidemic in Peru poses significant public health concerns, limited research exists on spatial modeling of the *Ae. aegypti* mosquito. This study addresses a critical gap by presenting the first comprehensive spatial analysis of its kind for Peru, offering valuable insights for decision-makers seeking to effectively address the disease burden. Accordingly, the present research aims to conduct a geospatial analysis to predict current and future distribution areas of the *Ae. aegypti* mosquito in Peru; additionally, it seeks to identify priority areas for targeted intervention strategies, directing resources and programs toward mitigating mosquito proliferation and implementing control measures.

## 2. Materials and Methods

### 2.1. Study Area

This research was conducted in Peru, located in South America, with coordinates ranging approximately from 0°2′ to 18°21′34″ latitude and 68°39′7″ to 81°20′13″ longitude. The Peruvian territory spans a total area of 1,285,215.9 km^2^ and exhibits diverse altitudes, ranging from below sea level to 6733 m above sea level (Figure 1). The country’s climate is highly varied due to its three primary longitudinal regions: (a) the arid coastal lowlands, (b) the Andean highlands, and (c) the Amazon rainforest with a tropical climate. Annual rainfall is distributed throughout the year in the Amazon region and the Andes, while precipitation patterns vary in the Andean highlands according to geographical characteristics (North, Central and South) [46].

### 2.2. Collection and Processing of the Presence Database

A total of 5617 reports of dengue documenting the presence of the *Ae. aegypti* species within the population centers of Peru were extracted from the official database maintained by the Peruvian Ministry of Health (https://www.datosabiertos.gob.pe/, accessed on 16 September 2023). This database comes from the National Epidemiology Network (RENACE), which is composed of 10,034 health facilities in Peru and is a historical series from 2000–2022. These reports are obtained from cases of dengue confirmed by chemical analysis in facilities in various regions of the country; those involved in the control and observations are health professionals trained in vector control (MINSA, 2023). The database was obtained from the portal in CVS format and was spatialized in a GIS software (v 3.36.3). Subsequently, a spatial filtering process was implemented to remove duplicate presence data sharing identical latitude and longitude coordinates; this process, aimed at mitigating potential sampling bias [47], resulted in 4853 presence records, which were utilized for subsequent model construction.

### 2.3. Variable Collection and Processing

We selected 20 variables in raster format to represent environmental factors (Table 1). The variables set including 19 bioclimatic variables corresponds to the whole standard WorldClim bioclimatic variables for WorldClim version 2, each with a spatial resolution of 30 arc seconds (~1 km^2^), obtained from the WorldClim 2.1 (www.worldclim.org/data/bioclim.html, accessed on 29 April 2025) platform (Table 1); here, the bioclimatic variables are averages for the years 1970–2000. Topographic variables were derived from a Digital Elevation Model (DEM) sourced from the Shuttle Radar Topography Mission (SRTM) through the United States Geological Survey (USGS) Geodata Portal (https://earthexplorer.usgs.gov/, accessed on 1 December 2023); subsequently, in a GIS software, the variables were clipped to the study area and converted to ASCII format [29,47]. To minimize the inference of autocorrelation between variables (20), the Pearson correlation method (Appendix A) was used through the “corrplot” package in R-4.2.3; variables with a correlation coefficient of ≥0.80 (Appendix A) were excluded (*) from further analysis to prevent multicollinearity issues [48,49,50], leaving 13 variables, of which 12 were bioclimatic and 1 was topographic.

### 2.4. Selection of Climate Models for Future Distribution

In order to project the future distribution of *Ae. aegypti* under climate change scenarios, a dataset comprising future climate predictions [30], based on Global Circulation Models (GCM) (EC-Earth3-Veg, HadGEM3-GC31-LL and MIROC6) of the CMIP6 obtained from the WorldClim 2.1 (https://www.worldclim.org/data/cmip6/cmip6_clim30s.html, accessed on 29 April 2025), was generated. Model selection was conducted for the timeframes of 2070 (period 2061–2080) and 2100 (period 2081–2100), considering two Shared Socio-economic Pathways (SSPs), taking into account two CO_2_ emission global scenarios, SSP 245 and SSP 585, and the moderate and highest values of anthropogenic radioactive force by year 2100 [51].

### 2.5. MaxENT Modeling

The modeling process involved integrating R version 4.2.3 and MaxENT software (v 3.36.3) with the use of “DISMO” and “Java” packages (https://cran.r-project.org); in this process, 75% of the presence data were randomly assigned for model training, with the remaining 25% left for validation [52]. The MaxENT model was configured with 10 simulation replications. Three key parameters were defined: 25 model training replications to ensure result robustness, a regularization multiplier of 1 to balance model complexity, and 10,000 backgrounds points for a robust reference database. These settings facilitate the optimal turning of the model, ensuring robust generalization and accuracy in predicting the distribution of *Ae. aegypti* [46,52]. The model’s output, ranging from 0 to 1, was categorized using the natural breaks cut-off method (Jenks), dividing it into 4 distinct ranges: inadequate (0–0.1), low suitability (0.1–0.3), moderate suitability (0.3–0.5) and high suitability (0.5–1.0) [53,54]

### 2.6. Model Validation

The model validation was performed using the Area Under the ROC Curve (AUC-ROC) as a metric to assess its accuracy, as is widely used in spatial modeling studies [46,53,54,55,56]. The AUC-ROC has the ability to evaluate the discriminative performance of predictive models using data such as presence and absence, as required by the MaxENT model [55]. AUC-ROC values range from 0 to 1, where *AUC* < 0.5 indicates random prediction, 0.5 ≤ *AUC* < 0.7 indicates poor model performance, 0.7 ≤ *AUC* ≤ 0.9 indicates moderate performance and *AUC* > 0.9 indicates high performance [15].$$AUC=\sum_{i=1}^{n-1}\frac{\left(X_{\left[i+1\right]}-X_{\left[i\right]}\right)\times \left(Y_{\left[i+1\right]}+Y_{\left[i\right]}\right)}{2}$$
where *X*_[*i* + 1]_ − *X*_[*i*]_ is the false positive rates of two consecutive points; *Y*_[*i* + 1]_ + *Y*_[*i*]_ is the corresponding true positive rates.

In addition, the *F*1-score and the Accuracy were utilized as evaluation metrics; the *F*1-score was measured using the harmonic mean of precision, positive predictive value and recall [57,58], and the Accuracy measures the proportion of correctly predicted outcomes among all observations, providing an overall assessment of the model’s predictive performance [54,59].$$Accuracy=\frac{TP}{TP+FP}$$
where *TP* and *FP* are true positive and false positive, respectively.$$F1-\mathrm{score}=\frac{2\times precision\times recall}{precision+recall}$$

The methodology developed in this research can be seen in the methodological flowchart (Figure 2), which shows the process from the collection of baseline data to the selection of topographic and bioclimatic variables, and current and future modeling with the MaxENT model.

## 3. Results

### 3.1. Statistical Metrics of the Distribution Probability Model

The current distribution model of *Ae. aegypti* in Perú, developed using the Maximum Entropy (MaxEnt) method, demonstrated significant accuracy, achieving an Area Under the Curve (AUC) of 0.91 on model training and 0.89 on testing (Figure 3A), and the variables that made the greatest contribution in the model were Elevation, Bio 6, Bio 3, Bio 12, and others (Figure 3B). The red line in the Figure 3A, represents the nondiscrimination line (or random reference line).

The statistical metrics reported by the model used in this study indicate a high reliability in the results generated, since the model obtained 0.9785 Precision, 0.8763 Recall, 0.9245 F1-score, 0.8751 Accuracy and 0.91 AUC, with these statistical metrics demonstrating that the topoclimatic model is reliable for the spatial distribution of *Ae. aegypti* in Peru (Table 2).

### 3.2. Probability of Areas for the Distribution of Ae. aegypti in Peru

The current (year 2023) mapping of the probability distribution areas of *Ae. aegypti* in Peru (Figure 4) indicates that 10.23% of Peru (132,053.96 km^2^) is highly suitable for the distribution of *Ae. aegypti* (Table 3). The departments with the largest territorial extension where there is the highest probability for the distribution of the species are the San Martin region (29,044.47 km^2^), Piura (19,434.85 km^2^), Loreto (15,598.88 km^2^), Lambayeque (10,470.23 km^2^), Cajamarca (7922.91 km^2^), Amazonas (7532.75 km^2^), and Cusco (7065.75 km^2^) (Figure 5). Based on the current modeling, the number of people possibly affected by *Ae. aegypti* was determined based on the high-probability class area obtained in this study. The results indicate that the departments of Piura, Lambayeque, Callao, San Martin and Lima have the highest numbers of people that could be affected by the adaptation of areas for the distribution of *Ae. aegypti* (Figure 5), with 1,126,356, 942,013, 771,973, 512,207 and 382,892 people possibly affected (Appendix A).

### 3.3. Probability of Future Areas for the Distribution of Ae. aegypti in Peru in Escenarios of Climate Changue

The future projection of the distribution of *Ae. aegypti* in Peru was conducted considering both an unfavorable scenario (SSP 585) and a favorable scenario (SSP 245) for the years 2070 (Appendix A) and 2100 (Appendix A). For the year 2022, the highly suitable area represents 10.23% of the Peruvian territory. By the year 2070, the HadGEM3-GC31-LL model exhibited the most pronounced increase in *Ae. aegypti* distribution, rising from 10.23% to 14.70% in the SSP 245 scenario and from 10.23% to 11.75% in the SSP 585 scenario. By the year 2100, in the SSP 245 scenario, the HadGEM3-GC31-LL model recorded a substantial increase in distribution extent, from 10.23% to 13.22%. Conversely, in the unfavorable SSP 585 scenario, the EARTH-VEG model demonstrated the highest increase, expanding from 10.23% to 10.69% (Table 4).

## 4. Discussion

This research evaluated the current and future distributions of *Ae. aegypti* in Peru, providing critical insights into the distribution dynamics of the dengue vector within the country. By identifying high-risk areas, this assessment establishes a foundational basis for the implementation of more targeted and effective control measures, ultimately safeguarding public health. The monitoring of the *Ae. aegypti* vector is emerging as a cornerstone in the prevention and management of dengue, a viral disease that imposes a significant global burden, particularly in tropical and subtropical regions. The substantial incidence rates and mortality associated with dengue highlight the urgent need for proactive measures in these areas [60].

The distribution of *Ae. aegypti* is primarily influenced by climatic factors that determine the survival and competence of this mosquito [61], as well as by social factors such as water storage practices in urban environments, which are crucial for vector development. Modeling distribution patterns based on social and environmental factors remains extremely challenging; therefore, several researchers have employed climatic variables as predictors for the spatial distribution of dengue fever [62]. However, Xu et al. [63] mentions that, in many cases, the future modeling of *Ae. aegypti* distribution is made complicated by the unavailability of up to date vector distribution data. This limitation was not present in this study, since MINSA provides an open database of the historical occurrence of dengue throughout Peru from 2000 to the present (2024).

There is a large number of studies on species distribution models (SDM) [64], all of which aim to explain, predict and project species distributions based on species occurrence data and environmental variables [26,65,66,67,68]. In this study we employed the MaxEnt model, a Species Distribution Model (SDM) sensitive to sampling bias. The MaxEnt model shows high predictive performance and comparable results for identifying test data under random and background weighting conditions, outperforming models such as the generalized linear model (GLM), gradient boosted model (GBM), and random forest (RF) [65]. GLM has been shown to achieve high predictive performance for test data, and exhibits lower specificity [69,70]. Furthermore, several studies indicate that GBM and RF are prone to overfitting training data [71] and the GLM overpredicts unsampled areas [72]; therefore, MaxEnt is capable of producing results that are both predictive (extrapolative) and complex (interpolative) [73,74,75], making it a practical method for addressing unbalanced and biased data in species distribution modeling approaches [26,43].

The performances of the Biomod2 model compared to MaxEnt for assessing prediction accuracy are similar, as both models can produce accurate predictions with appropriate occurrence inputs and simulation iterations. However, Biomod2 requires a longer run time and lower data processing power [56]. In this study, to avoid sampling bias and improve the accuracy of prediction results attributed to MaxEnt, a database of 10,000 randomly generated pseudo-absences was created [71] using the “smaplerandom” function of the “Raster” package in Rstudio v. 4.2.3.

The current distribution model indicates that *Ae. aegypti* is primarily concentrated in the coastal and Amazonian regions of our country, consistent with previous studies [76,77,78]. These findings align with research demonstrating that the spatial distribution of the *Ae. aegypti* predominantly occurs in tropical and subtropical areas. As noted by [79], the distribution pattern is attributed to favorable climatic conditions that support the mosquito’s life cycle and facilitate the transmission of diseases such as dengue. Consistently warm temperatures, coupled with high humidity, provide favorable conditions for the egg hatching and larval development of *Ae. aegypti* mosquitoes. Moreover, abundant rainfall in these regions creates numerous breeding sites in both naturally formed and artificial containers [80], emphasizing that the primary climatic factors influencing dengue transmission include temperature, precipitation, and relative humidity.

In urban settings, anthropogenic landscape features can influence mosquito dispersal patterns [81]. Yeo et al. [82] found that, in urban agglomerations, particularly in densely populated residential areas, the gene flow rates of vector species are higher. In Peru, it is projected that by 2035, the urban population will increase to 31.3 million people, representing 81.8% of the total population [83]. This would likely enhance the dispersal of *Ae. aegypti*, while climate variability across complex urban environments could further provide favorable conditions for the vector [81].

It was observed that coastal regions such as Lambayeque, La Libertad and Piura exhibit the largest areas with high suitability for the distribution of *Ae. aegypti*. Despite the scarce rainy seasons in these regions, this phenomenon is explained by the widespread practice of water storage in containers within urban areas, which provides favorable conditions for vector breeding and reproduction [26]. This issue constitutes a persistent global challenge, as dengue vectors can withstand high temperatures and reproduce efficiently in clean water stored in containers.

At the territorial level, the variation in species distribution areas obtained in the results is due to the different changes projected by each climate model. Moreover, it is recognized that climate change will alter Andean biomes, particularly affecting the páramos of Bocayá (Colombia), Azuay and Loja (Ecuador), and Piura and Cajamarca (Peru) [84]. These alterations are primarily due to rising temperatures, which, as previously discussed, are conducive to the breeding, development, and propagation of *Ae. aegypti* [85].

The main results indicate that, under the 2070 year scenario, both the most favorable scenario (SSPs 245) and the unfavorable scenario (SSPs 585) highlight the HadGEM3-GC31-LL model, showing a significant increase in areas with high distribution suitability for the species, with 14.70% and 11.75%, respectively. By the year 2100, the EC-Earth-Veg model shows the most substantial increase in areas in the high-suitability class for Peru, with 11.77% and 10.69% in both SSPs 245 and 585, respectively. These results are similar to those of other authors, who also showed, using future projection with climate models of the CMIP6 complex, that the density [86] and area suitability [87] of *Ae. aegypti* will increase with climate change due to temperature increases [88].

Future projection results using the EC-Earth-Veg and HadGEM3-GC31-LL models are valid and consistent. The EC-Earth-Veg model couples soil biophysical parameters to generate vegetation parameters, and interacts with atmospheric variables such as humidity [89], demonstrating strong performance compared to the other models in the CMIP6 ensemble [90]. These parameters are critical for the distribution of *Ae. aegypti* in Peru, as they provide the necessary conditions for the survival and propagation of the species. Additionally, the HadGEM3-GC31-LL model shows good performance in estimating precipitation and temperature [91,92], key variables for predicting vector distribution; both models have been widely utilized in biodiversity and conservation studies due to their reliability and robust results in predicting the distribution of flora and fauna species [93,94,95].

The propagation of *Ae. aegypti* is related to socioeconomic variables because these determine the environmental and structural conditions that favor vector development [96]. The absence of continuous access to drinking water in low-income populations forces domestic storage in exposed containers that establish optimal habitats for mosquito oviposition [97]. Furthermore, inadequate sanitary infrastructure, such as poor sewage systems and inefficient solid waste management, increases the amount of anthropogenic microhabitats within stagnant water, increasing the survival rate of larval stages of *Ae. Aegypti* [98], and high population density and informal urbanization generate a high availability of human hosts, reducing the flight distance required for feeding and thus intensifying viral transmission [99].

On the other hand, higher population density has been associated with a stronger relationship between temperature and *Ae. aegypti* incidence, compared to areas of low population density [100]. This is because unplanned urbanization facilitates closer proximity between the population and larval habitats, encourages inadequate water storage practices that produce breeding sites, and promotes high human mobility, thereby contributing to the greater dissemination of the dengue virus [101]. Additionally, the relationship between precipitation and vector population size is not linear, as it depends more on the intensity and frequency of rainfall events [102]. Intense rainfall in natural and artificial habitats can increase the suitability of breeding sites, trigger the hatching of *Ae. aegypti* eggs and promote larval development [103].

The findings of this research are particularly relevant given the alarming dengue situation in Peru, as indicated by reports issued by the Ministry of Health in 2024. As of March 2024, Peru has reported a staggering 34,042 cases of dengue, representing a significant increase of 131% compared to the same period last year; in addition, the number of fatalities rose to 44. Coastal departments, including La Libertad (6148 cases), Piura (5275 cases), Ica (4645 cases), Ancash (3766 cases) and Lima (2899 cases), have witnessed a higher prevalence of dengue. These reports contrast sharply with the findings of this research, which identify coastal departments such as Piura, Lambayeque and La Libertad as departments with high and very high distributions of *Ae. aegypti*.

The results of this research will be available for use by the relevant entities in charge of entomological and epidemiological interventions in areas where *Ae. aegypti* has not yet been recorded, but which have been predicted with the model generated in this study.

## Figures and Tables

**Figure 1 insects-16-00487-f001:**
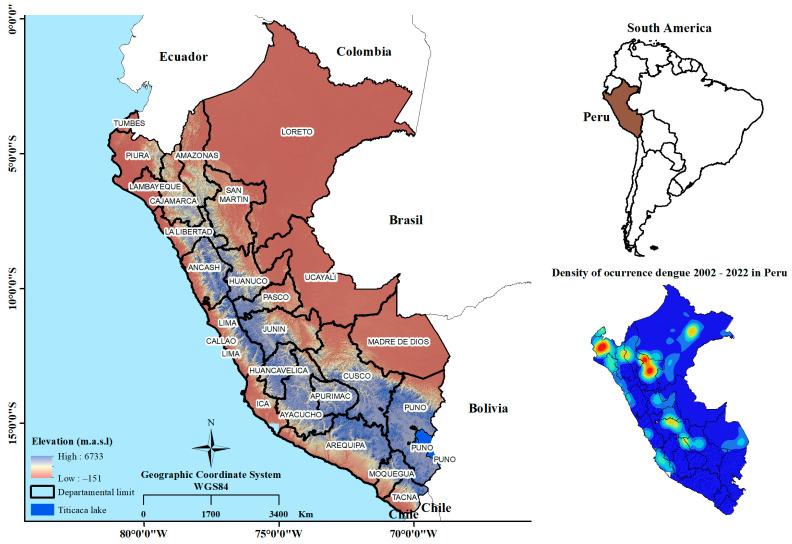
Location of the study area.

**Figure 2 insects-16-00487-f002:**
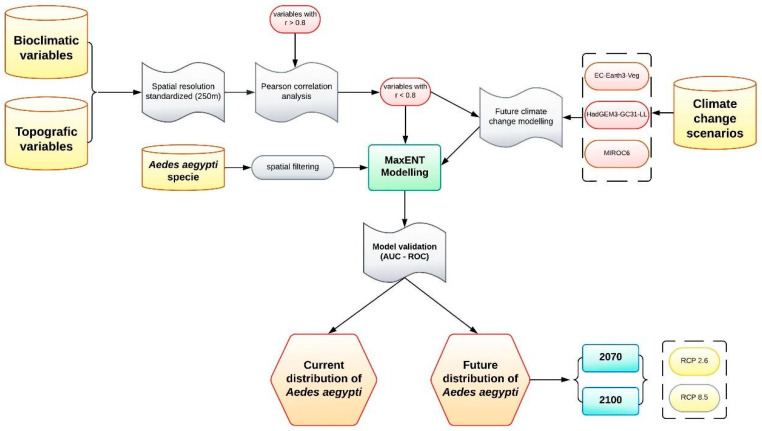
Methodological process flowchart.

**Figure 3 insects-16-00487-f003:**
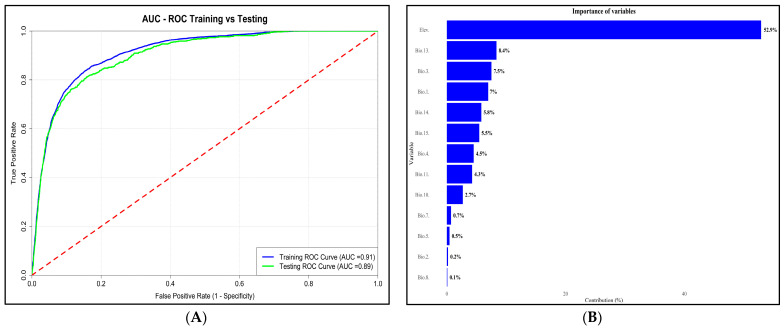
Distribution probability model. (**A**) Area Under the Receiver Operating Characteristic Curve (AUC-ROC). (**B**) Importance of variables integrated in the model.

**Figure 4 insects-16-00487-f004:**
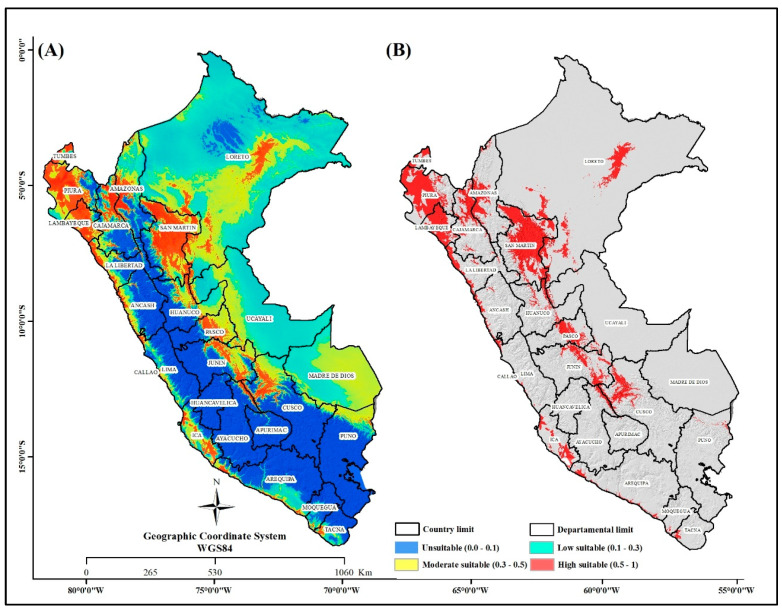
Distribution area of *Ae. aegypti* in Peru. (**A**) Distribution levels for Peru. (**B**) Highly suitable area for *Ae. aegypti* distribution in Peru.

**Figure 5 insects-16-00487-f005:**
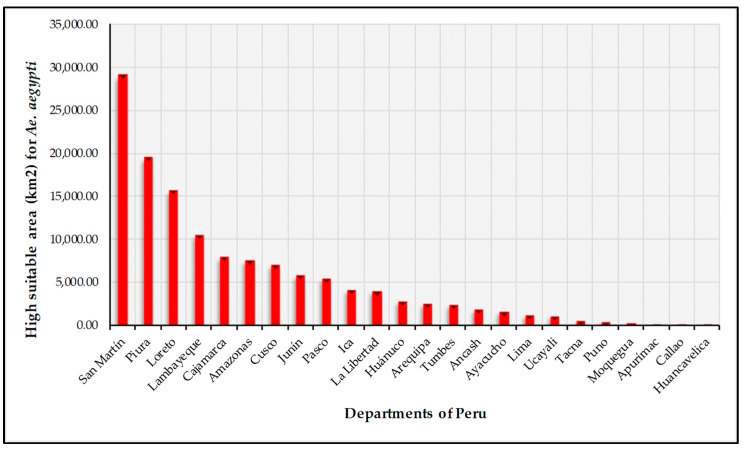
Area (km^2^) of high suitability for *Ae. aegypti* in each department of Peru.

**Table 1 insects-16-00487-t001:** Variables used for modeling.

Type	Variables	Description	Source
Environmental variables	bio01	Annual Mean Temperature	WorldClim
	bio02	Mean Diurnal Range (Mean of monthly (max temp–min temp))	
	bio03	Isothermality (BIO2/BIO7) (×100)	
	bio04	Temperature Seasonality (standard deviation ×100)	
	bio05	Max Temperature of Warmest Month	
	bio06	Min Temperature of Coldest Month *	
	bio07	Temperature Annual Range (BIO5-BIO6)	
	bio08	Mean Temperature of Wettest Quarter	
	bio09	Mean Temperature of Driest Quarter *	
	bio10	Mean Temperature of Warmest Quarter	
	bio11	Mean Temperature of Coldest Quarter	
	bio12	Annual Precipitation *	
	bio13	Precipitation of Wettest Month	
	bio14	Precipitation of Driest Month	
	bio15	Precipitation Seasonality (Coefficient of Variación)	
	bio16	Precipitation of Wettest Quarter *	
	bio17	Precipitation of Driest Quarter *	
	bio18	Precipitation of Warmest Quarter *	
	bio19	Precipitation of Coldest Quarter *	
Topographic variables	DEM	Elevation	SRTM

Note: (*) variables removed from the initial set (correlation ≥ 0.80).

**Table 2 insects-16-00487-t002:** Statistical metrics of the distribution probability model.

Statistical Metrics	Precision	Recall	F1-Score	Accuracy	AUC
MaxEnt	0.9785	0.8763	0.9245	0.8751	0.91

**Table 3 insects-16-00487-t003:** Probability classes for *Ae. aegypti* distribution in Peru.

Probability Class Distribution	Area	
	Km^2^	%
Unsuitable	416,352.51	32.27
Low suitability	436,850.80	33.85
Moderate suitability	305,253.82	23.65
High suitability	132,053.96	10.23
Total	1,290,511.09	100

**Table 4 insects-16-00487-t004:** Quantification of areas of distribution for the future projection of *Ae. aegypti* according to specific climate change scenarios.

Global Climate Model	Year	SSPs	High Suitable		Diference
			Km	%	%
Actuality	2022		132,053.96	10.23	10.23
EC-Earth3-Veg	2070	245	145,759.56	11.29	1.06
HadGEM3-GC31-LL			189,715.13	14.70	4.47
MIROC6			163,304.19	12.65	2.42
EC-Earth3-Veg		585	121,197.74	9.39	−0.84
HadGEM3-GC31-LL			151,664.19	11.75	1.52
MIROC6			139,376.75	10.8	0.57
EC-Earth3-Veg	2100	245	151,899.69	11.77	1.54
HadGEM3-GC31-LL			170,608.69	13.22	2.99
MIROC6			146,618.13	11.36	1.13
EC-Earth3-Veg		585	138,003.37	10.69	0.46
HadGEM3-GC31-LL			135,695.46	10.51	0.28
MIROC6			136,953.93	10.61	0.38

## Data Availability

The original contributions presented in this study are included in the article and Appendix A. Further inquiries can be directed to the corresponding author.

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
