# Peer review of "Current and Future Spatial Distribution of the Aedes aegypti in Peru Based on Topoclimatic Analysis and Climate Change Scenarios"

_insects, 2025, doi:10.3390/insects16050487_

Round 1
Reviewer 1 Report
Comments and Suggestions for Authors
The manuscript describes a project to model current and future distribution of the dengue vector Ades aegypti in Peru under different climate change scenarios. This work is important given the rapid expansion of dengue cases in Peru and the dramatic outbreak in Central and South American in 2024. The MaxEnt algorithm is commonly used for mosquito species distribution modelling and is a good choice in this work. The model uses an extensive database of mosquito surveillance. Overall, the manuscript describes a useful and interesting modelling effort and will be a valuable addition to the literature.
On the other hand, there are some problems that need to be improved, particularly with the writing and organization. Also, as is often the case with mosquito modelling papers, the description and consideration of mosquito biology and human/vector interactions are a bit weak. The authors refer to Ae. aegypti presence and dengue cases almost interchangeably which is not correct. While vector presence is of course a prerequisite for transmission, many regions of the world maintain robust vector populations but dengue transmission is rate. The authors need to tidy up the langue and distinguish between mosquito distribution and dengue.
The discussion of dengue is missing key information on dengue serotypes and the issues of asymptomatic cases as well as overall limitations in case data due to lack of detection and underreporting. Furthermore, there is no mention of extrinsic incubation period, the time necessary for the virus to move from an infected bloodmeal in the mosquito gut to the salivary glands, where it can be passed to another human. It is not necessary to incorporate EIP into the model, which is tricky, but it should at least be acknowledged as this period is highly influenced by temperature. It is possible to have a cooler climate that allows the vector to survive but does not allow it to live long enough to become infective. There is a substantial literature on this topic that should be discussed in both the introduction and the discussion.
Specific comments:
Abstract: The first sentence is far too long and should be divided into at least two shorted sentences.
Introduction: This section needs to significant revision. The authors go into excessive detail on some topics and omit other important topics (mentioned above). Paragraph 1 is fine with a few changes noted below. Paragraph 2 should combine lines 52 to 68 but needs to focus on Ae. aegypti, not dengue. I suggest the authors discuss larval habitat requirements a bit more and explain how piped water influences mosquito abundance. More information on this species' temperature preferences would be good too. Omit the reference to Aedes albopictus, as that is not the species being modelled here. In paragraph 3, the authors should add information about the different serotypes, the frequency of asymptomatic infections and the link between previous infections and the hemorrhagic form. What information is available on the serotypes currently circulating in Peru? The rest of the introduction is good but again, too long and detailed. Some places to reduce the text are listed below.
Line 45 - Change 'The mosquito to 'Mosquitoes'.
Lines 50-51 - Omit sentence. It is interesting but tangential to the work being presented here.
Lines 51-52 - Sometimes the authors refer to the vector as Ae. aegypti (correct) and sometimes as A. aegypti (incorrect).
Lives 62-65 - Omit sentence.
Lines 72-73 - Omit sentence.
Lines 79-82 - Omit sentence.
Lines 92-106 - Reduce the list of districts and regions. Maybe a map would be better.
Line 128 - Remove the mention of edaphological conditions. As a container breeder, Ae. aegypti is unlikely to be influenced by soil in the way that other species are.
Lines 129-131 - Maybe I am confused, but I read the paper cited and did not find that it established an association between dengue incidence and environmental factors. It was focused on the vector only.
Methods: I am not a modeler, so I am not the best judge of the thoroughness of the methods section. Having said that, it looks good to me! One question related to the mosquito presence database. On lines 192-195, the authors refer to reports confirmed by chemical analysis. I am confused. I thought the data were mosquito counts, so what chemical analysis are they refering to. If it is dengue cases, that needs to be explained clearly and in more detail than chemical analysis. There are lots of ways to confirm dengue infections and your results will be influenced by which method you use.
Lines 107-126 - Suggest combining the two paragraphs and reducing the text, particularly references to the global statistics.
Line 144 - I think there is a word missing here.
Results: Good overall but way too detailed. Some of the text information belongs better in a table and some of the tables below in a supplemental section. Also, I'm a bit confused about the factor elevation, which will clearly be associated with the climate variables, particularly temperature. Again, not a modeler so maybe I am completely missing something here but how does the model deal with that?
Section 3.2 paragraph 1 and 2 should be combined. Cut lines 287 starting with 'furthermore...' as the figure 4 shows this already. Lines 298 and 299 are duplicates but should just be combined into a single population figure. Table 5 should be included in a supplemental section.
Figures 5 and 6 show very similar patterns. I'm wondering if the later projections could be in a supplemental section, as they don't really add much and are easily confused with Figure 5, which is really the star of this manuscript.
Discussion: Good overall, but too long and slightly repetitive. I suggest removing the paragraph on lines 336-340 and the paragraph on lines 434 to 439, as these topics have already been discussed. Similarly, the paragraph on lines 395 to 406 could be omitted or considerably shortened and added to another paragraph in the discussion. Finally, the conclusion section seems unnecessary to me, as this manuscript really wasn't about public health interventions. Maybe it could be edited and incorporated into the Discussion.
Comments on the Quality of English LanguageThe manuscript is well-written overall. Throughout the manuscript, the authors use the citation number directly in the text to cite particular studies (e.g. Previous research, as demonstrated by [46] have identified...) Is that the normal practice for this journal? I thought is was still necessary to refer to the reference by author name in this case (e.g. Previous research, as demonstrated by Jacome et al. [46] have identified..)
Author Response
"Please see the attachment."

Reviewer 2 Report
Comments and Suggestions for Authors
Importance of the Study
The study is highly relevant for monitoring and controlling Aedes aegypti in Peru, especially in the context of climate change. The article uses modeling based on the MaxEnt algorithm to predict the current and future distribution of the mosquito, providing valuable information for control and prevention strategies against dengue. The findings indicate that the vector will continue to expand its distribution in the country, particularly in already vulnerable regions such as San Martín, Piura, and Loreto. This highlights the need for more targeted preventive actions to minimize disease spread.
Points That Need to Be Corrected
- Correction of Scientific Formatting
- Scientific names, such as Aedes aegypti, Aedes albopictus, and others, should always be in italics throughout the text.
- Clarity and Cohesion in Writing
- The article contains long and complex sentences that make reading difficult. Some passages could be rewritten for better clarity.
- Certain sections repeat information about the climatic impacts on mosquito distribution, making the reading redundant.
- Presentation of Results
- The results section presents many numerical values that could be better visualized through more intuitive graphs and tables.
- Some tables could be reorganized to facilitate comparison between the different climate scenarios analyzed.
- Discussion and Contextualization
- Although the discussion is well-supported, it would be useful to include more comparisons with similar studies conducted in other South American countries to contextualize the findings within a broader scenario.
- The relationship between socioeconomic variables and the spread of *Aedes aegypti* could be further explored, considering factors such as urbanization and sanitation infrastructure.
- English Language Review
- Some sentences contain minor grammatical and cohesion errors that may affect the text’s fluency. A review by a native English speaker or a scientific writing expert would help ensure greater clarity.
Conclusion
The article is scientifically relevant and presents well-structured data, but it requires revisions in formatting, organization of result presentation, and writing adjustments to improve clarity and cohesion. After these corrections, it has great potential to be published in journals focused on medical entomology, climate change, and epidemiology.
Comments on the Quality of English Language
The writing is good, but it can be improved.
Author Response
"Please see the attachment."
